# Seeing through the forest: The gaze path to purchase

**Bridget K. Behe**[1]*, **Patricia T. Huddleston**[2], **Kevin L. Childs**[3], **Jiaoping Chen**[4], **Iago S. Muraro**[2]

**1** Department of Horticulture, Michigan State University, East Lansing, Michigan, United States of America, **2** Department of Advertising & Public Relations, College of Communication Arts & Sciences, Michigan State University, East Lansing, Michigan, United States of America, **3** Department of Plant Biology, Michigan State University, East Lansing, Michigan, United States of America, **4** Eli Broad College of Business, Michigan State University, East Lansing, Michigan, United States of America

* behe@msu.edu

## Abstract

Eye tracking studies have analyzed the relationship between visual attention to point of purchase marketing elements (price, signage, etc.) and purchase intention. Our study is the first to investigate the relationship between the gaze sequence in which consumers view a display (including gaze aversion away from products) and the influence of consumer (top down) characteristics on product choice. We conducted an in-lab 3 (display size: large, moderate, small) X 2 (price: sale, non-sale) within-subject experiment with 92 persons. After viewing the displays, subjects completed an online survey to provide demographic data, self-reported and actual product knowledge, and past purchase information. We employed a random forest machine learning approach via R software to analyze all possible three-unit subsequences of gaze fixations. Models comparing multiclass F1-macro score and F1-micro score of product choice were analyzed. Gaze sequence models that included gaze aversion more accurately predicted product choice in a lab setting for more complex displays. Inclusion of consumer characteristics generally improved model predictive F1-macro and F1-micro scores for less complex displays with fewer plant sizes Consumer attributes that helped improve model prediction performance were product expertise, ethnicity, and previous plant purchases.

## Introduction

Most consumer product choices (>90%) are made at the point of purchase (POP) [1]. Retailers have a variety of tools (signs, labels, display fixtures, merchandise arrangement, etc.) for organizing the store environment to capture consumer attention and motivate consumer choice. Consumer packaged goods companies invested approximately $29.7 billion in shopper marketing in 2016, accounting for more than 13% of their marketing budget [2]. Marketing organizations perceive this investment to be effective, as indicated by 73% of surveyed executives who rated in-store marketing as very or quite useful [3], making the retail space a critical location for purchase decisions.

**Data Availability Statement:** The data and code are held in OSF and available at https://osf.io/hykwu/?view_only=e8f1350ee39e41d39af59155be96e517.

**Funding:** The study was funded by the United States Department of Agriculture – Federal State Marketing Improvement Program Grant 16FSMIPMI0007 which was awarded to bkb, pth, and klc. Their website is located at https://www.ams.usda.gov/services/grants/fsmip. Additional funding was provided by the Horticultural Research Institute grant #6012448 and awarded to bkb, pth, and klc. Their website is located at https://www.hriresearch.org/.

**Competing interests:** The authors have declared that no competing interests exist.

When shopping, consumers identify visual cues then cognitively process those cues to make a purchase decision [4, 5]. Visual attention, which is requisite for purchase, is the process by which stimuli are selected and integrated for cognitive function allocated to the task at hand. Portable and affordable eye tracking (ET) equipment has created a reliable, objective mechanism to study visual cue selection leading to purchase. Past ET research has shown that single cues such as signage [6, 7], pricing [8], packaging [9], brand [10, 11], and product display [12] influence visual attention and purchase intention. Yet, consumers often ignore some visual cues. For example, Hendrickson and Ailawadi [13] observed that in a store setting, shoppers do not look up unless they are "way finding" and store signs are noticed only by about 3% of shoppers. While visual time in product selection has been investigated, the order in which consumers access visual cues (e.g., gaze sequences) has not been widely studied.

Gaze sequence, or the progression of visual cues to which consumers attend, is not random, but task specific [14]. Some ET research has studied the areas at which consumers look prior to product choice, but a breakdown of the gaze sequence merits more investigation. In an in-store setting, we assume that when given a shopping task consumer search is purposeful [14] and that there might be gaze sequences that are predictive of choice. That is, not just what a consumer views or how often an item is viewed, but the order in which components of a display are viewed may be informative of consumer choice. Additionally, by solely focusing on consumer visual attention to products and POP communication tools, previous studies have not investigated the possible relationship between looking away from product visual cues and purchase choice. The gaze aversion literature [15–17] suggests that humans may sometimes need to look away from a main visual target to facilitate information processing. Since the range of product choice and the number of visual cues at the POP can be overwhelming at times, it seems plausible to conjecture that a similar mechanism may affect consumers who seek to alleviate their mental discomfort of "choice overload" [18]. Thus, the purpose of our study is to investigate whether a relationship exists between subcomponents of consumer gaze sequence and product choice for a minimally packaged, unbranded product, and to understand whether looking away, which we term Look Away to Decide (LATD), plays a role in a consumer's gaze sequence on their path to product choice.

## Study significance and literature gaps

The majority of ET studies focus on branded and/or fast-moving packaged consumer goods displayed on shelves [10, 19, 20], with products in identical packaging size and product quality assessed by reading labels or examining packaging. Our study focuses on non-branded, similar but non-uniform (non-identical) products (live plants) where quality assessment may require product scrutiny. In the next section, we review studies that have focused on aspects of gaze sequence, such as gaze aversion and central gaze bias. It is important to note, however, that few of the studies investigated the gaze paths in which visual cues were accessed and no studies investigated whether that relationship predicted product choice. Our study integrates extant literature findings and expands them by documenting the patterns in the cue selection process that leads to a product choice. In our analysis of visual behavior, we add a new and heretofore unexamined viewing behavior, LATD. The findings of our study have both theoretical and practical implications. From a theoretical standpoint, linking features within consumer gaze sequence to product choice aids in our understanding of cognitive processing of visual cues. Commonalities and differences in gaze sequences can lend insight into product choice by pinpointing cues that are selected (vs. ignored). For managers, the ability to pinpoint one or more gaze paths that lead to purchase intention will enable retailers to reimagine the environment in which merchandise is presented, (e.g., signage/pricing placement and display spacing). Our

goal is to better understand subcomponents of the gaze sequence that are associated with product selection by consumers.

## Review of literature

**Decision-making cues.**   Eye-tracking has been utilized in hypothetical consumer studies related to product choice, largely through laboratory investigations of attendance and non-attendance to package, label, or menu cues. For example, Ballco et al [21] demonstrated nutritional claims were influential in a hypothetical discrete choice experiment. Tórtora and colleagues [22] found nutritional warnings attracted attention and required less time and fewer fixations to process compared to facts on a front panel. Additionally, Otterbring and Shams [23] used eye-tracking while showing a video to subjects in a lab setting to find that prior exposure to a seemingly healthy consumer produced greater visual attention towards products perceived to be healthy and eliciting healthier cereal choices. That study extended prior work showing student subjects who viewed unhealthy body types depicted on a menu spent more visual time viewing healthy food options [24].

Visual attention time correlates positively with purchase, such that increased fixation time on certain product attributes indicates an increased purchase likelihood [25–29]. Gidlöf et al. [27] reported, "the very act of looking longer or repeatedly at a package, for any reason, makes it more likely the product will be bought." Similarly, Scarpi et al. [20] found that fixation time for prototypical products increased purchase intention. Khachatryan et al. [30] found subjects who scored high on the buying impulsiveness scale fixated less on POP information and more on the product (live plants). The studies cited above focus on how consumers view a display (e.g., time fixating on the product, specific attributes, or display features), rather than the sequence in which these cues are viewed.

**Top-down and bottom-up cues.**   Attention to visual cues implies an awareness of the stimuli in the conscious mind, which is driven by both top-down and bottom-up processes [14, 31]. In the visual cognition literature, bottom-up processes refer to stimulus driven attention [32] or low-level stimulus features that capture attention. Research on bottom up processes finds that visual attention is driven by salient stimuli features (e.g. contrast, color) and is mostly involuntary and reactive [33]. Top down processes refer to the conscious experience of visual stimuli [34], are context dependent [35] with visual attention being allocated based on the nature of the task [33, 35]. That is, attention is drawn to relevant areas based on the task at hand (e.g. selecting a blue flower). The visual cognition literature suggests that in the absence of a task, bottom-up processes influence visual attention, but once the task tells us what to look for, it changes the way we look at stimuli [36]. In other words, in the absence of a task, visual attention might be driven by the most salient feature of the stimuli, i.e. driven by a bottom-up process, but once a task is assigned visual attention will be directed toward those features that fulfill the task.

Wedel and Pieters [14] apply the findings of visual cognition literature to visual attention to marketing. They posit that (marketing) stimuli salience and informativeness act in combination to affect attention. Aligned with the visual cognition literature, Wedel and Pieters [14] propose that bottom-up processes operate when consumers attend to salient features such as color or contrast and this attention is mainly involuntary. Top-down processes, as identified by Wedel and Pieters [14] refer to perceptions of marketing stimuli that could be influenced by consumer demographics, brand familiarity, expertise, involvement and attitudes or goals. Unlike tasks assigned in the visual cognition literature which have a correct/incorrect outcome [33, 35] marketing study tasks are usually open ended (which brand would you buy) with no right or wrong answer. Outcomes are determined by consumer factors. For this study, we are

interested in analyzing which, if any, consumer characteristics influence gaze pattern for plant displays.

Additionally, prior studies have shown that consumers who are more knowledgeable (have greater expertise) about a product make purchase decisions differently than less knowledgeable consumers [37]. Expertise arises from training, practice, or time spent learning about a particular topic [38, 39]. Alba and Hutchinson [40] and Shanteau [41] reported that consumers with high product expertise were more selective of the information they examined prior to making a choice since they had a better understanding of what product attributes should be examined. Expertise has been linked to visual attention as well [42].

**Gaze sequence.** Several ET studies provide researchers with a starting point for understanding the role of gaze paths in purchase intention and product choice. In an early ET study conducted in a retail context, Russo and Leclerc [43] characterized four stages of a consumer choice process. They documented a sequence of visual cue selection, followed by deliberate and effortful brand elaboration, then by final fixations and post-announcement (choice) fixations. Clement [19] expanded the findings of Russo and Leclerc [43] by using gaze paths to identify several decision phases: pre-attention, succeeded attention, tipping point, physical action, semantic information process, and post-purchase phases. Çöltekin, et al. [44] reported that geography experts (vs. novices) had shorter, and thus more efficient, gaze sequences when completing the task of finding a point on a map. Josephson and Holmes [45] had mixed results in their effort to identify gaze patterns when subjects viewed a webpage. Drusch et al. [46] also investigated scanpaths for webpages and identified two groups, from 91 subjects, with similar scanpaths using Hausdorff distance metrics. The empirical contribution of these studies is the documentation of a multi-stage gaze or search strategy that precedes purchase decision; however, none of these studies specified particular gaze sequences that guided this process and led to product choice.

An ET study of online information search patterns demonstrated that customers limited their attention to approximately three attributes for one product [47]. A key finding was the identification of a gaze sequence in which a preferred alternative was used as the basis, or anchor, for comparing other alternatives. Shi et al.'s [47] model contributes to the understanding of gaze paths or patterns by illustrating how this phenomenon is empirically linked to a preferred alternative, with that alternative becoming the basis for comparison of other alternatives. Thus, early in the search, process consumers appear to identify an initial (product) candidate and make iterative comparisons back to that primary choice. This would make the first visual point of contact as the one that establishes the bar or standard for others to meet or exceed. Onuma et al. [48] further showed that the second look at a chosen item was longer than the first look, which they posit was evidence for encoding information sequentially with a comparison phase in choice. Rebollar et al. [49] identified a gaze sequence (referred to as scanpath by the authors) of chocolate bar attributes as focusing initially on the bar name (the most prominent feature), followed by the brand. However, these studies neither compare sequences of fixations beyond the first few nor do they examine how fixation patterns can be informative of purchase decision.

Literature linking gaze sequence to consumer information processing and decision-making is scant and disparate. The published literature involving product choice utilizing ET technology includes studies mainly focused on first, last, central, or important areas of interest (AOIs) rather than gaze sequences. Atalay et al. [50] demonstrated the central gaze effect where branded products in the center of a vertical display had greater visual activity and were more likely chosen. Armel et al. [51] showed that the likelihood of selecting the product on the left (in a binary choice scenario) increased with longer gaze time. Reutskaja et al. [52] conducted a series of simulated choice studies on a computer screen and incorporated eye-tracking. They

showed that as the number of product options (snack items) increased, consumer slightly increased the number of alternatives evaluated. Furthermore, products positioned centrally in the vertical display had more first fixations especially when the number of choices in a set increased, consistent with Atalay et al. [50]. Bialokva et al. [53] conducted lab and in store studies for product choice in a supermarket. They found that product brand and flavor were the key drivers of visual interest. Huddleston et al. [54] identified five studies that used packaged goods to link visual measures to actual product choice in the retail environment [7, 9, 19, 43, 55], but none involved gaze sequence analysis.

Related to the Shi et al. [47] study are two ET studies that indicate a central gaze bias tendency and identification of different viewing behaviors for horizontal versus vertical displays. Atalay et al.'s [50] studies investigated how horizontal location on a webpage affects choice, finding that brands in the center were more frequently chosen, received more eye fixations, and were looked at longer. Additionally, consumers were more likely to fixate on the centrally located brand in the last few seconds of gaze duration; these central fixations increased the likelihood of choosing a brand located in the center. They concluded that a centrally located option in a product display is preferred and garners more attention. However, the central position did not have an impact on the inferences that consumers made about the fictitious brands (e.g., amount of market share) or memory of which brand they chose [50]. This suggests that a centrally located position does not influence consumer product judgments. Comparing horizontal versus vertical display types, Deng et al. [56] discovered that consumers purchased higher quantities and greater varieties from horizontal displays; horizontal displays increased processing fluency; thus consumers perceived that horizontal assortments had a higher level of variety; and when faced with time constraints, consumers spent more time processing horizontal displays and fixated on a larger number of items. For a menu selection, related somewhat to product choice, Yang [57] showed that the scanpath of subjects reading a restaurant menu consisted of two vertical (top to bottom) scans; this finding was counter to the almost circular viewing path the restaurant industry hypothesized.

Lahey and Oxley [58] acknowledged the great utility that visual metrics have in behavioral economic studies and called specifically for analyzing the scan path or gaze sequence. Still, within the realm of product choice, scant work has been published implementing gaze sequence analysis. Thus, there is a need for more investigation in the gaze sequence related to product choice. Based on these findings, central position and horizontal displays appear to enjoy an advantage in stimulating consumer choice. Our study, while informed by the central gaze theorem, differs from previous work in that we focus on horizontal versus vertical displays and, rather than using identical branded packages, our stimuli are unbranded plants that vary slightly in appearance.

Thus, we pose our first two research questions:

RQ1: Are consumer gaze sequence paths predictive of product choice?

RQ2: Do consumer characteristics (top-down factors) enhance the predictive accuracy of gaze paths?

**Gaze aversion (look away to decide).** When faced with a plethora of visual cues (e.g., signs, labels, product assortment), consumers must somehow visually select and then cognitively process these cues to arrive at a decision. Glenberg et al. [17] opined, "when engaged in difficult cognitive activity, we close our eyes or look at the sky to suppress the environment's control over cognition (p. 651)." Psychology literature suggests that humans are predisposed to avert their gaze directed at others (gaze aversion) when faced with difficult decisions or

complex mental tasks as an attempt to facilitate cognition. Several researchers [15–17] suggested that in face-to-face interactions, people needed to avert their gaze when answering difficult questions. Gaze aversion helps individuals avoid processing unnecessary visual cues and, consequently, enhances their cognitive control and improves their performance when answering questions [16, 17]. One study [59] pinpointed an adverse effect of gaze aversion when they found that interview answers were less reliable in conversational interviews when there was gaze aversion.

The gaze aversion literature has focused on face-to-face human interaction, not consumer viewing patterns in the context of product choice. Nonetheless, we believe that these findings can be extrapolated to a shopping environment because, similar to face-to-face interactions, consumers in a shopping context are inundated with an excess of visual cues and may need "visual space" to cognitively process this information. In a shopping environment, there are often many extraneous visual cues that do not aid in decision making, and it is possible that consumers may engage in LATD to facilitate their decision-making process.

The authors conceptualize LATD as the direction of consumers' central gaze (gaze fixations) toward visual cues away from the choice task at hand. Therefore, LATD is not related to peripheral vision. Gaze centrality is the area to which the eyes are fixed and indicate selective attention directed to a visual cue [50, 60]. On the other hand, peripheral vision is the visual area lying outside the central gaze, or narrow foveal angle, with a limited perception of shapes and colors [61, 62]. Scant literature about peripheral vision and product choice provide evidence that consumers can engage their vision peripherally to eliminate non-relevant visual cues from active consideration [61]. The opposite happens when consumers engage in LATD, they redirect their central gaze either to visually process non-relevant cues instead, such as the store ceiling, the floor, or a product display background to reduce or eliminate stimulus input for the moment.

Thus, we pose our third research question:

RQ3: Does LATD improve predictive accuracy in a gaze path sequence to product choice?

## Methods

### Open science statement

Data, analytic code, stimuli, survey instrument, and supplemental information (Appendix A depicting how 3-mers were developed) can be found online in the Open Science Framework (OSF) website: https://osf.io/hykwu/?view_only=5e36084775af40218c6a1617b3337e3b

### Stimuli

The study protocol and questionnaire were approved by the Michigan State University Institutional Review Board (17–458) which included approval and use of a written consent form. We conducted a 3 (display size) X 2 (price) within-participants experimental design in May 2017 (examples of the display designs are depicted in Figs 1–3). The six table-top displays showed three varieties of flowering plants [large (24 plants), moderate (12 plants), small (6 plants)] at two price levels (sale, non-sale). The sign displayed either the sale (low) price ($1.99) or the non-sale (high) price ($3.99) using black Calibri font (size 80 for non-price wording and 40 for price information). Text and spacing were identical across all displays. Displays were isolated from others with black cloth that extended fully around the sides of each display. Plants occupied approximately 0.12 m$^2$, 0.25 m$^2$, and 0.45 m$^2$ of display space for the 6, 12, and 24-plant designs, respectively. Each display provided about 0.46 m$^2$ of display area. All participants

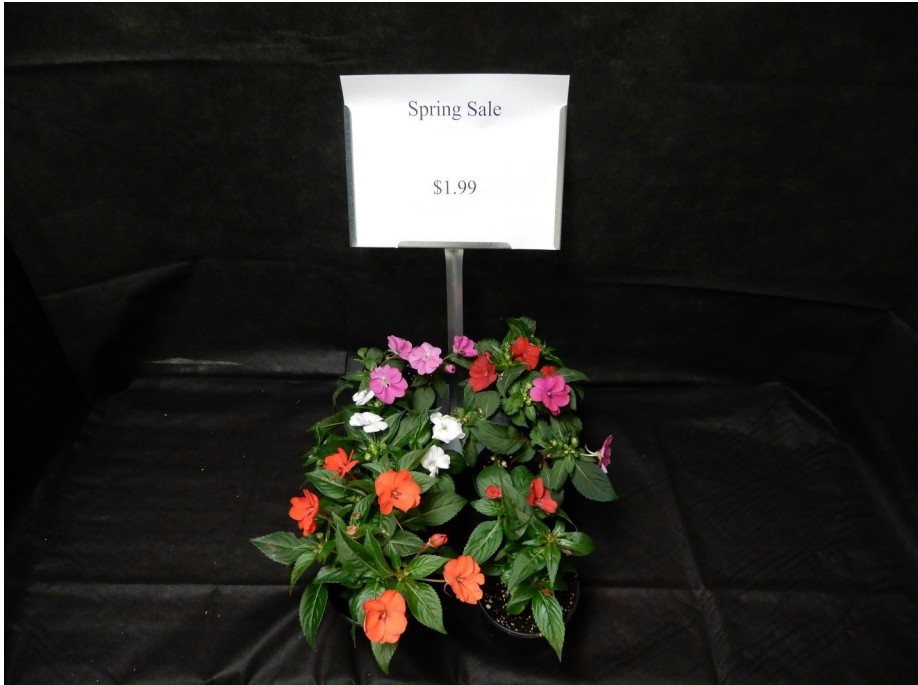

**Fig 1. Example of six-plant display with low price sign.**

were exposed to a total of six plant display conditions, and subjects could view only one display at a time and began the study at successively different displays to reduce order bias. Each

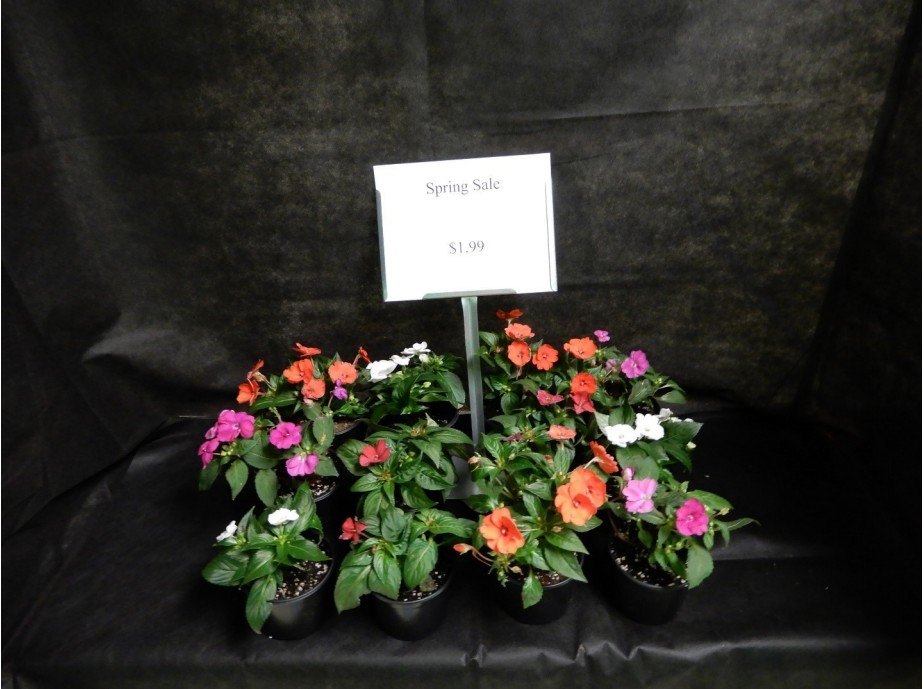

**Fig 2. Example of 12-plant display with low price sign.**

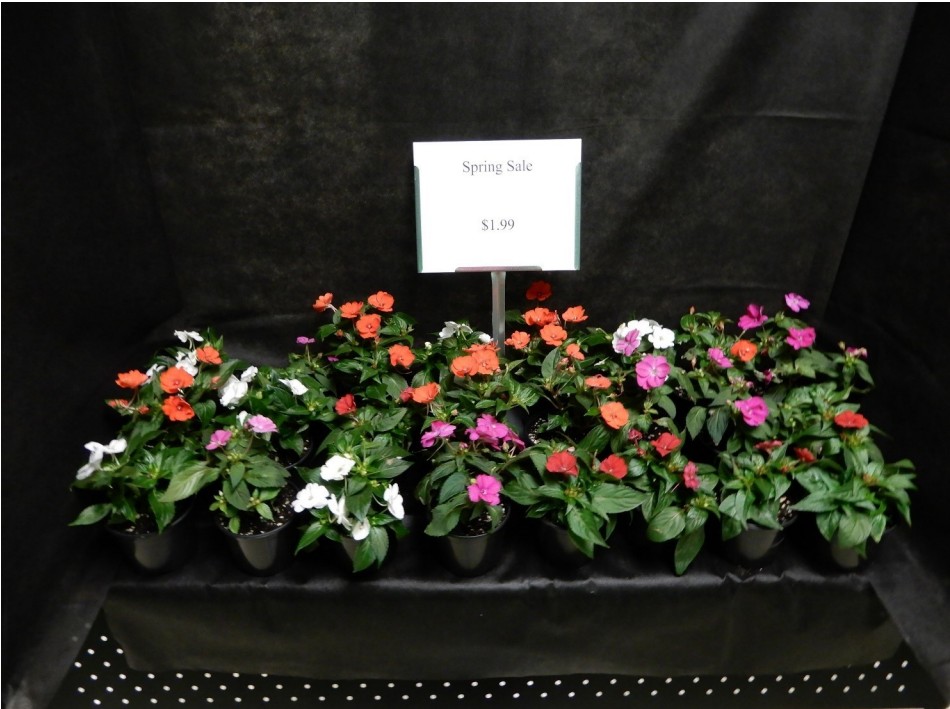

**Fig 3. Example of 24-plant display with low price sign.**

subject proceeded from one display to the next unaccompanied using a progressive start to the first display with no other participants viewing displays. Two researchers were in the same room, but out of the visual line of sight while subjects participated in the study.

## Sample and protocol

In the lab at a midwestern university, subjects were asked to view each of the six displays. Their task was to the one product (plant) from each display that they would be most likely to purchase and verbally indicate how likely they would be to purchase the plant using the 11-point Juster scale [63]. Alternatively, subjects could indicate they would not buy any of the plants in a display, reflecting a 0 on the Juster scale. After viewing all plant displays, subjects completed an online survey that measured their level of both self-reported and actual product expertise (following [42]) as well as their purchase history in the six months prior to the study. To measure purchase history, participants answered the following question: "How much did you spend (in total) on gardening supplies and plants (excluding mechanical equipment like mowers and tillers)?" Response options included "$0", "$1 to $24", "$25 to $49", and then increased in $50 increments up to "$500 or more". The survey also collected their demographic information.

Subjects [$n$ = 92, mean (standard deviation) age = 35.3 (12.9) years; females, 69.6%; identified as white, 68.5%; identified as Asian, 17.4%; identified as African American, 4.4%; and, mean income = $73,294 ($50,412)] were recruited through an online panel maintained by researchers (panel comprised of both students and non-students) at a large Midwestern university and paid $25 for their participation. After completing the informed consent process, subjects were fitted with Tobii Pro Glasses 2 eye tracking glasses (with a sampling frequency of 50 Hz, 20 ms sampling interval, and binocular accuracy of 0.62˚) and given three practice

**Table 1. Two-way repeated measures ANOVA on likelihood to buy.**

| Factor | Df | F-value | p-value |
|---|---|---|---|
| Display_size | 2 | 9.551 | 0.000114* |
| Price | 1 | 15.63 | 0.000152* |
| Display_size * Price | 2 | 7.524 | 0.000725* |

rounds with chocolate bars to familiarize them with the product choice task and Juster scale. They were instructed first to point to the product (candy bar or plant) they were most likely to buy, and then to verbally state their likelihood of buying that item using whole numbers from 0 to 10 [63] with a score of 0 indicating that the subject chose to not make a purchase. The displays were set in a random order in the lab, but subjects viewed the randomized displays in sequential order, with each subject starting in a successive sequence. Table 1 shows a two-way ANOVA with repeated measures to evaluate the effect of different display sizes (small, moderate and large) with various prices (sale vs not-sale) on the likelihood to buy. Two main effects (display size and price) are significant, which indicates the display sizes and price both affect consumers' likelihood to buy. There is also a statistically significant interaction between display sizes and price on the likelihood to buy, F-value = 7.524, p<0.0001. Fig 4 indicates that participants' buying likelihood on average are greater in a display under the sale price. Besides, the sale price, displays with a large (24 plants) or small (6 plants) number of product options tend to have greater mean likelihood to buy, compared to the moderate display size.

## Data handling and analysis

We digitally created areas of interest (AOIs) around each plant using Tobii Pro Lab software (version 1.73) and assigned those AOIs a letter (A–F in the six-plant display, A–L in the 12-plant display, and A–X in the 24-plant display), with the informational sign denoted as "Y"

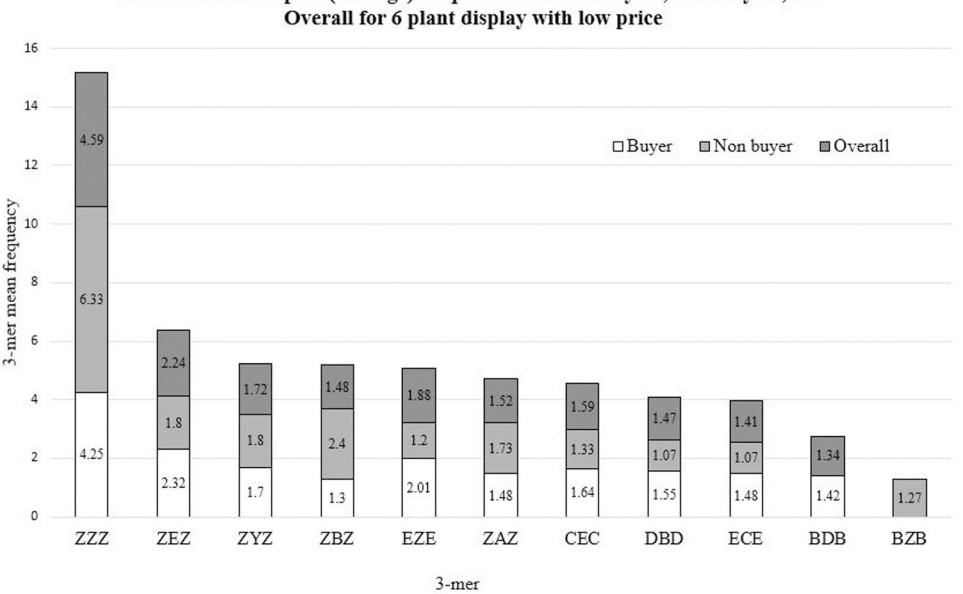

**Fig 4. Average likelihood to buy score with 95% confidence intervals over three display sizes (small, moderate and large) and price (sale and not-sale).** The black square represents "not-sale" group, and the red square represents "sale" group.

and gazes off the sign or plant (gaze aversion) denoted as "Z". The raw gaze data for each participant, available in 20 ms intervals, was exported, then reduced to just the gaze changes between AOIs. Thus, for each subject at each display, we created a gaze sequence file showing the progression of gaze events throughout the display, ending when the subject made a choice. For example, the gaze sequence with looking-away fixation of one participant viewing the six-plant display was "ZZYZYCECACACZYBYZZBYB" until verbal choice. To interpret this gaze sequence, the subject looked away (Z) six times, looked at the sign (Y) four times and looked at plant C four times, plant E once and plant B three times. Similarly, the gaze sequence without looking-away fixation would be "YYCECACACYBYBYB".

A custom Python program was prepared to parse the gaze sequence files. For each gaze sequence, all possible subsets of consecutive sequential fixations (called k-mers, where k is the number of sequential fixations) were identified and subjected to further analyses. For example, when k is equal to 3 (3-mer), relative frequencies of consecutive subsequences with length 3 (ZZY, ZYZ, YZY, ZYC, YCE, CEC, ECA, etc. for the first gaze sequence example above) are extracted as features contained in each gaze sequence with look-away fixations (see Appendix A for a detailed example). The use of k-mers to break down the gaze fixation sequences is inspired by k-mer analysis of nucleic acid sequences in bioinformatics [64, 65]. Single fixations or sequences of two fixations (2-mers) hold very little information about participants' sequential gaze. However, as the number of fixations in a sequence increased to 4-mers and above, the proportion of participants sharing an identical gaze sequence decreased drastically, thus causing an analytical problem of finding relevant patterns in the data. This dilemma is also known as sparse feature matrix [66]. For that reason, we examined gaze patterns as sets of 3-mer fixations. In this study, 3-mers retain quantitative data regarding the total number of fixations on each object in the display, but they also preserve information about the order of fixations between the objects in the display. Furthermore, these 3-mers intuitively often contain gaze patterns employed by the test subject. In the gaze sequence above, CAC, ACA, YBY and BYB suggest a comparative thought process by the subject. We found 3-mers to be both interpretable as well as sufficiently abundant for the analyses in this project. The next step was to investigate whether gaze 3-mers would be predictive of product choice.

Two machine learning classifiers, Random forests (RF) and Support Vector Machine (SVM) were used to predict final plant purchase choice (outcome variable) using participants' gaze patterns (the relative frequency of 3-mers) and survey data. The RF classifier is an ensemble of multiple decision tree models that select the best set of predictors to partition the data into smaller sets with high within-group homogeneity in respect to the dependent variable [66]. The SVM is another popular machine learning tool for classification tasks, which constructs a set of hyperplanes in a high-dimensional space to categorize various groups. We selected RF and SVM machine learning approaches because they are both well suited for classifying high-dimensional data sets such as the present one, which would be impractical in the context of traditional linear models due to parameter estimation problems. In this study, we conducted the RF classification via the *randomForest* R package [67] and SVM classifier via the *e1071* R package.

The RF implementation followed Sundararajan et al.'s [68] approach. First, we used a 4:1 split ratio to randomly divided the data into two non-overlapping sets: (1) a training set and (2) an external testing set. Using the training set only, we employed a fivefold cross validation to identify an optimal model. The *randomForest* options ntree (5, 10, 15, 25, 50, 75, 100, 250, 500) and nodesize (0.5, 1, 2) were varied, and the best combination of ntree and nodesize values, as determined by the average classification performance for 5-fold internal test sets within the training set, were retained to construct the final model [66, 69]. All other *randomForest* parameters were set to their default options. The final model was then applied to the external

testing set [70]. Meanwhile, we applied similar procedures for SVM classifier with a radial basis kernel. Two hyperparameters within the radial basis kernel, the *gamma* option (0.1, 0.3, 0.5, 0.7, 0.9, 1, 2, 10) and *cost* option (0.01, 0.1, 1, 5) were tuned.

The best average classification performance was indicated by: (1) Confusion entropy (CEN), which is a misclassification measure ranging from 0 (optimal classification performance) to 1 (the lowest classification performance); (2) Overall predictive accuracy (OA), which is the ratio between the number of correctly-predicted cases relative to the total number of cases; (3) Average predictive accuracy (AA), which is an auxiliary measure of accuracy that evaluates the average model performance in respect to each category of the dependent variable (see [71, 72]) for a review of the performance indicators); (4) F1-macro score, which is a F1 score (the harmonic mean of the precision and recall) with averaging precision and recall of each individual class; and (5) F1-micro score, which is calculated from individual true positives, true negatives, false positives and false negatives. In the multi-class classification task, F1-micro is also the classifier's overall accuracy. Reported results are based on the external testing set, a standard practice to increase the generalizability of research findings [70].

Our final models focused on predicting consumer choice for plants selected by at least 10 consumers (top two plants in six-plant displays and top three plants in the 12- and 24-plant displays) and for the no-plant choice. Consequently, the effective sample size ranged from 43 to 92 across displays. Specifically, displays 1–6 had sample sizes with 92, 76, 54, 77, 43 and 56 for classification tasks, respectively. Because eye-tracking technology captures a tremendous amount of data per participant, the effective sample size range was appropriate for analytical purposes with the lower bound (43) being slightly below the average sample size (46.59) of 75 other similar eye tracking studies reported in two meta-analytical articles [73, 74]. The decision to model the most commonly selected plants and the no-plant choice is logically supported by our intention to understand the most prevailing consumer choices as opposed to focusing on rarely selected plants; because this focus helps develop parsimonious RF classifiers; and because our use of independent data subsets for the RF classifier optimization [68] means that a sparse outcome would be undesirable as it could yield an over-specified model with low external validity.

We collected consumer attributes in an online survey following display viewing and product choice. Thus, we constructed four models with various features: Model 1 considers frequencies of consecutive 3-mers without looking away fixations from the gaze sequence (denoted as "3-mer-without-LATD"). Model 3 considers frequencies of consecutive 3mers with looking away fixations (denoted as "3-mer-with-LATD"). Model 2 and Model 4 included consumer attributes from the online survey in addition to the features used in Model 1 and Model 3, respectively. However, not all the features may have been relevant for predicting product choice. Given each model setting, variable selection was performed before modeling by using ANOVA F-test for every numerical feature, in order to extract those with variations among various final purchasing decision/plant. Feature reduction helped to remove irrelevant features, reduce the dimensionality of feature space and reduce over fitting of the trained model.

## Results

We examined the most common gaze sequences (3-mers) by plant display. Fig 5 displays the average frequency in which the top ten 3-mers occurred across the participants' gaze sequences for one display (6 plants, sale price, as shown in Fig 1). This information shows that, irrespective of the number of plants in the display or the plant price, looking away from the products (denoted by the letter Z) is extremely frequent. For instance, for the display with 6 plants, sale

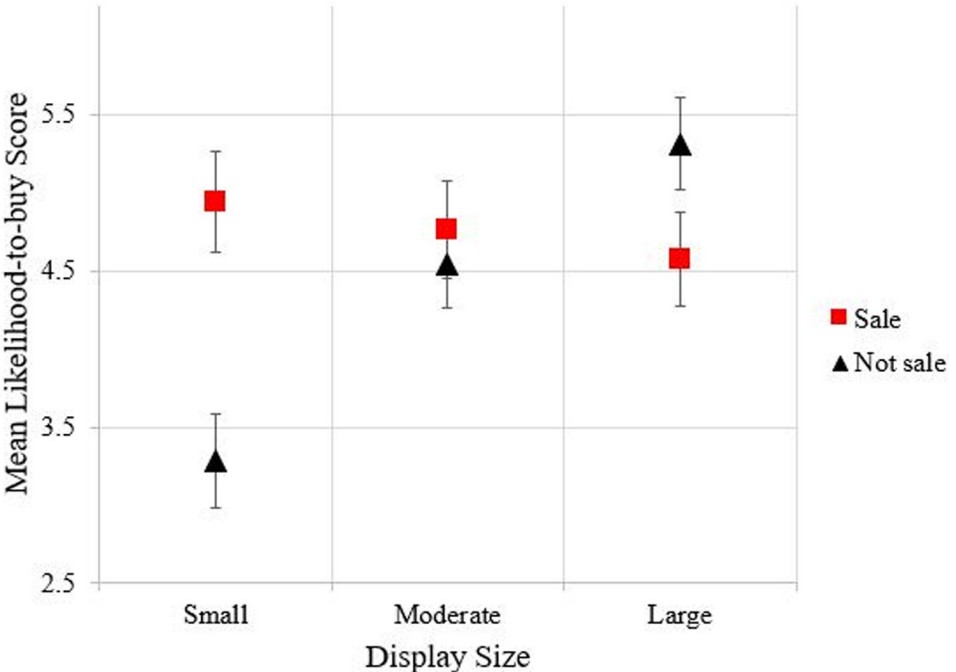

**Fig 5. Average frequency for top ten 3-mers in display with six plants at the low price.**

price, the 3-mer ZYZ (i.e., looking away, looking at the price signage, and looking away again) occurred 1.72 times for each participant overall, while it occurred 1.8 times for the non-buyers and 1.7 times for subjects who elected to purchase a plant from that display. The percentage of look-away fixations was statistically above expectation (**Table 2**).

Next, we evaluated the predictive performance of the eight final RF and SVM classifiers. Only three variables from the consumer attributes contributed meaningfully to the models' predictive accuracy: expertise, ethnicity (68.5%), and the prior purchase of annual plants (58.7%). The zero-order correlations between three consumer attributes are shown in **Table 3**.

First, we examined the performance of model 1. For instance, for a 6-plant, sale-price display, using 3-mers alone resulted in a low entropy (CEN) of 0.3 with 72% accurate predictions (OA), which is ~2.2 times more accurate than randomly guessing product choice (prediction baseline = 1/6). The average accuracy (AA) of model predictions by choice (top two plants and no-plant choice) was 44%. Overall, the predictive performance of all models could be improved for at least one of the metrics by including consumer attributes.

Moreover, we compared the final RF classifier performance when gaze sequences and LATD were employed with and without consumer attributes (models 3 and 4). Table 4 shows the CEN and overall and average accuracies for each model by display type. While CEN was identical for one model (6-plant, non-sale display), it was lower (representing less error) for four of the five remaining models, indicating that the addition of consumer attributes (model

**Table 2. Two-way repeated measures ANOVA on percentage of looking-away fixations.**

| Factor | Df | F-value | p-value |
|---|---|---|---|
| Display_size | 2 | 30.75 | 3.13e-12* |
| Price | 1 | 0.948 | 0.333 |
| Display_size * Price | 2 | 2.641 | 0.074 |

Table 3. Correlation between the included consumer attributes.

| | Expertise | Ethnicity | Prior Plant Purchase |
|---|---|---|---|
| Expertise | 1.00 | 0.35 | 0.47 |
| Ethnicity | | 1.00 | 0.38 |
| Prior Plant Purchase | | | 1.00 |

4) enhanced the model's performance. For two of the six models, overall accuracy was identical with and without the addition of consumer attributes. However, overall accuracy was improved with the addition of consumer attributes for three of the models while it was lower for the model predicting choice in the largest display with a non-sale price. Average accuracy was similar for one model (six-plant, non-sale display) but higher for three of the remaining models (see Table 5). Thus, the preponderance of evidence leads us to conclude that model performance is enhanced by the inclusion of consumer attributes (model 4). The model associated with the sale price and large display has the highest predictive performance across all models when consumer attributes are included, resulting in a very low CEN of 0.08, and 75% of accurately predicted product choices (overall and by choice category), which is approximately three times better relative to the random prediction for a 24-plant display (4%).

Lastly, we compared the final resulting RF classifier performance when gaze sequences and consumer attributes are employed with and without LATD information (models 2 and 4). Table 6 shows the results of the model performance with and without LATD gazes. Generally, CEN was lower or slightly lower with LATDs for five of six displays. The exception was for the moderate-sized 12-plant display with a sale price. Overall accuracy was greater or equal when including LATD for three of the six models, and average accuracy had a similar result. Taking all the parameters into account, lower error and similar or higher accuracy when LATD events were included were achieved for five of the six displays with the LATD information (model 4) compared to without (model 2). Our highest predictive accuracy was for the smallest display with a low price (78%).

## Discussion and conclusion

Our first research question centered on whether features contained within gaze sequences are predictive of product choice. These results provide evidence that breaking down a consumer's

Table 4. Comparison of predictive accuracy across all four random forest models.

| Display Number | Number of Plants | Price | Confusion Entropy (CEN) | | | | Overall Accuracy (OA) | | | | Average Accuracy (AA) | | | |
|---|---|---|---|---|---|---|---|---|---|---|---|---|---|---|
| | | | Model 1 | Model 2 | Model 3 | Model 4 | Model 1 | Model 2 | Model 3 | Model 4 | Model 1 | Model 2 | Model 3 | Model 4 |
| 1 | 6 | Low | 0.3 | 0.38 | 0.29 | **0.27** | 0.72 | 0.67 | **0.78** | 0.72 | 0.44 | 0.42 | **0.56** | 0.44 |
| 2 | 6 | High | 0.64 | **0.63** | 0.64 | **0.63** | **0.53** | **0.53** | **0.53** | **0.53** | 0.55 | **0.57** | 0.55 | **0.57** |
| 3 | 12 | Low | 0.43 | 0.43 | **0.25** | 0.36 | 0.55 | 0.55 | **0.73** | 0.55 | 0.54 | 0.58 | **0.71** | 0.54 |
| 4 | 12 | High | 0.42 | 0.4 | 0.36 | **0.35** | 0.47 | 0.53 | 0.53 | **0.60** | 0.35 | 0.4 | 0.43 | **0.53** |
| 5 | 24 | Low | 0.46 | 0.39 | 0.39 | **0.08** | 0.38 | 0.25 | 0.38 | **0.75** | 0.38 | 0.25 | 0.38 | **0.75** |
| 6 | 24 | High | 0.33 | **0.28** | 0.54 | 0.45 | 0.55 | **0.73** | 0.45 | 0.36 | 0.54 | **0.71** | 0.46 | 0.38 |

Model 1: 3mer-without-LATD;

Model 2: 3mer-without-LATD + consumer attributes;

Model 3: 3mer-with-LATD;

Model 4: 3mer-with-LATD + consumer attributes.

Note: The numerically superior result in each model (lower CEN, higher OA, and higher AA) is displayed in bold.

**Table 5. Comparison of predictive accuracy across for random forest and SVM models.**

| Classifier | Number of Plants | Price | F1-macro score (f1_macro) | | | | F1-micro score (Overall Accuracy) (f1_micro/OA) | | | |
|---|---|---|---|---|---|---|---|---|---|---|
| | | | Model 1 | Model 2 | Model 3 | Model 4 | Model 1 | Model 2 | Model 3 | Model 4 |
| RF | 6 | Low | 0.25 | **0.32** | 0.24 | 0.30 | 0.72 | **0.78** | 0.67 | 0.72 |
| SVM | 6 | Low | 0.20 | 0.20 | 0.20 | 0.24 | 0.67 | 0.67 | 0.67 | 0.67 |
| RF | 6 | High | **0.28** | **0.28** | **0.28** | **0.28** | **0.53** | **0.53** | **0.53** | **0.53** |
| SVM | 6 | High | 0.16 | 0.16 | 0.16 | 0.16 | 0.47 | 0.47 | 0.47 | 0.47 |
| RF | 12 | Low | 0.29 | **0.36** | 0.28 | 0.29 | 0.55 | **0.73** | 0.55 | 0.55 |
| SVM | 12 | Low | 0.17 | **0.40** | 0.34 | 0.34 | 0.36 | **0.73** | 0.64 | 0.64 |
| RF | 12 | High | 0.18 | 0.26 | 0.20 | **0.30** | 0.47 | 0.53 | 0.53 | **0.60** |
| SVM | 12 | High | 0.10 | 0.10 | 0.10 | 0.10 | 0.33 | 0.33 | 0.33 | 0.33 |
| RF | 24 | Low | 0.16 | 0.19 | 0.11 | **0.39** | 0.38 | 0.38 | 0.25 | **0.75** |
| SVM | 24 | Low | 0.18 | 0.28 | **0.40** | 0.38 | 0.38 | 0.50 | **0.75** | **0.75** |
| RF | 24 | High | 0.33 | 0.25 | **0.39** | 0.19 | 0.55 | 0.45 | **0.73** | 0.36 |
| SVM | 24 | High | 0.31 | 0.08 | 0.14 | 0.04 | 0.55 | 0.18 | 0.27 | 0.09 |

Model predictors:

Model 1: 3mer-without-LATD;

Model 2: 3mer-without-LATD + consumer attributes;

Model 3: 3mer-with-LATD;

Model 4: 3mer-with-LATD + consumer attributes.

Note: The numerically superior result in each display under either F1-macro or F1-micro score. The higher F1-macro score and the higher F1-micro score (is equivalent to overall accuracy in classification tasks) are displayed in bold.

gaze path into short segments (3-mers) of visual fixations can help us to predict product choice. Chance of randomly guessing would be 33% for 6 plant displays (due to 3-class classification), and 25% for all other displays (due to 4-class classification). Our models indicate that consumers' gaze sequence alone could accurately predict choice from 38% (large display, sale price) to 72% (small display, non-sale price) of consumers, which represents an improvement ranging from four to about nine times relative to the prediction baseline. Russo and Leclerc [43] and Clement [19] documented distinct consumer fixation patterns across the multiple phases of in-store purchase decision process. Our findings advance that knowledge by providing an indication that consumers' fixation sequences are predictive of choice. However, our results indicate that the predictive power associated with consumer gaze sequence can be improved by the addition of consumer attributes or LATD information in various settings.

**Table 6. Models with the best performance.**

| Number of Plants | Price | % of LATD fixation on average among participants | Model 1 3mer-without-LATD | Model 2 3mer-without-LATD + Consumer attributes | Model 3 3mer-with-LATD | Model 4 3mer-with-LATD + Consumer attributes |
|---|---|---|---|---|---|---|
| 6 | Low | 38% | | ■ • | | |
| 6 | High | 40% | ■ • | ■ • | ■ • | ■ • |
| 12 | Low | 35% | | ■ • | | |
| 12 | High | 33% | | | | ■ • |
| 24 | Low | 31% | | | ■ • | • |
| 24 | High | 33% | | | ■ • | |

The square (■) represents the model with the best **F1-macro score** among 8 model settings (4 models * 2 machine learning method). The circle (•) represents the model with the best **F1-micro score** (is equivalent to **overall accuracy** in this classification task) among 8 model settings.

Our second research question focused on whether consumer characteristics help to improve the predictive accuracy of gaze paths in product choice. While gaze sequence alone predicts purchase intention with high accuracy (up to 78% overall accuracy in the sale price, 6-plant display), we were able to improve the overall accuracy for simpler displays when consumer attributes (top-down factors) were included. Three top-down factors (expertise, ethnicity and prior plant purchase) enhanced predictive accuracy. Purchase and plant-related activities in the U.S. historically has been an activity participated in largely by more affluent white consumers. This is evidenced in nearly three decades of research conducted by the National Gardening Association (see https://gardenresearch.com/) also documented in Dennis and Behe [75]. However, more recent NGA studies show the gap between white and persons from other ethnic backgrounds closing as participation among non-white consumers increases.

These results also aligns with the findings of Joo et al. [42] that expertise influenced time of visual attention to both intrinsic and extrinsic cues. This effect was evident especially for the larger, non-sale priced display, perhaps suggesting that more cognitive resources are needed when faced with more financially risky decisions (i.e., higher price). Similar to the present study, other physiological measures have been related to purchase choice [68, 76]. Ravaja et al. [76] observed a price effect with more electroencephalogram (EEG) readings, which measure brain activity, for lower-priced products which may have been an indication of further consideration; while there was less brain activity for higher-priced products which may have been dismissive as a potential choice, indicating that brain activity is influenced by product price. Sundararajan et al. [68] used EEG readings to predict choice of 181 students in 10 dichotomous food choices with an average peak accuracy of 84%. Our study used visual attention as the physiological measure. When comparing our results to the aforementioned studies, in a purchase decision task with more choices (24 here versus Sundararajan et al.'s [68] two choice study), our predictive accuracy improved when using gaze patterns in combination with consumer attributes.

In answer to research question 3 as to whether gaze aversion plays a role in product choice, we found that (1) look-aways (LATD) are included in ~70% of the 10 most frequent gaze sequence 3-mers, and that (2) the inclusion of LATD reduced error in 5 of 6 models and produced a somewhat higher predictive accuracy. Therefore, it seems that looking away from product displays is not only a frequent behavior at the POP but is also a meaningful predictor of consumer choice. Other studies acknowledge the importance of looking at the product [48] and the time spent viewing the product [25–29]. Here, we interpret the abundance of look-aways in the gaze paths to suggest that the look-away is also important for cognition and allows reflection, which helps confirm a product choice. The predictive value of including LATDs was greater for displays with more plants and higher prices. This suggests that having a sale price or lower price option does not require as much cognitive effort and that the look-away gazes for these lower priced options do not appear to help finalize the decision to the same extent as higher-priced or bigger displays. This also confirms the importance of bottom-up factors in the choice process.

For the first time to our knowledge, the need to look away to inform the purchase choice (LATD) is documented. Computer data must be processed in order to create output. Here, we see the visual behavioral corresponding to when possible cognition is likely occurring in the look-away gazes and their inclusion in modeling choice has been shown to improve predictive accuracy in product choice. One managerial implication of our findings is that in creating a merchandise display, retailers should add neutral space where consumers' visual attention can focus while higher cognitive processes are involved in the decision-making process.

There are several limitations to laboratory studies including the present study. First, subjects were asked to imagine a purchase decision. Conducting the study in situ introduces the

real distractions and motivations for shopping, which would enhance the external validity of a study. Furthermore, the type of product (live plants) may not appeal to all subjects who agreed to participate in the study. Additional studies with related packaged goods (e.g. fertilizer, pest controls, etc.) could provide additional insight with regard to visual assessment of alternatives prior to choice or selection.

This exploratory study invites many avenues for future research. Only one product type (live plants) was investigated; thus, future studies should bring in other product types and include branded merchandise and consumer packaged goods. Future studies should explore the effect of larger and more elaborate displays on gaze sequence and LATD. Additionally, more extensive research is needed to understand the differential functions and the interplay of centrally (LATD) and peripherally [61] gazing at non-relevant visual cues.

## Acknowledgments

The authors acknowledge the technical support of Lynne Sage and Jie Wang.

## Author Contributions

**Conceptualization:** Bridget K. Behe, Patricia T. Huddleston, Kevin L. Childs.

**Data curation:** Kevin L. Childs, Jiaoping Chen.

**Formal analysis:** Kevin L. Childs, Jiaoping Chen.

**Funding acquisition:** Bridget K. Behe, Patricia T. Huddleston, Kevin L. Childs.

**Investigation:** Bridget K. Behe, Patricia T. Huddleston.

**Methodology:** Bridget K. Behe, Patricia T. Huddleston, Kevin L. Childs, Jiaoping Chen.

**Project administration:** Bridget K. Behe.

**Resources:** Bridget K. Behe, Patricia T. Huddleston, Kevin L. Childs, Jiaoping Chen.

**Software:** Kevin L. Childs, Jiaoping Chen.

**Supervision:** Bridget K. Behe, Patricia T. Huddleston, Kevin L. Childs.

**Validation:** Bridget K. Behe, Patricia T. Huddleston, Kevin L. Childs, Jiaoping Chen, Iago S. Muraro.

**Visualization:** Bridget K. Behe, Patricia T. Huddleston, Kevin L. Childs, Jiaoping Chen, Iago S. Muraro.

**Writing – original draft:** Bridget K. Behe, Patricia T. Huddleston, Kevin L. Childs, Jiaoping Chen, Iago S. Muraro.

**Writing – review & editing:** Bridget K. Behe, Patricia T. Huddleston, Kevin L. Childs, Jiaoping Chen, Iago S. Muraro.

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
