## [Decision Letter · Decision Letter 0]

13 Jul 2020

PONE-D-20-14107

Seeing through the forest: The gaze path to purchase

PLOS ONE

Dear Dr. Behe,

Thank you for submitting your manuscript to PLOS ONE. After careful consideration, we feel that it has merit but does not fully meet PLOS ONE’s publication criteria as it currently stands. Therefore, we invite you to submit a revised version of the manuscript that addresses the points raised during the review process.

Reviewers provided a deep and detailed evaluation of the paper. They highlight several critical points and offer some useful comments that might help improving it. In particular, you need to be more convincing about the reliability of the analysis, the applied methods and their choice, and the data quality. 

We look forward to receiving your revised manuscript.

Kind regards,

Claudio Soregaroli

Academic Editor

PLOS ONE

Journal Requirements:

2. Please modify the title to ensure that it is meeting PLOS’ guidelines (https://journals.plos.org/plosone/s/submission-guidelines#loc-title). In particular, the title should be "specific, descriptive, concise, and comprehensible to readers outside the field" and in this case it is not informative and specific about your study's scope and methodology

Reviewers' comments:

Reviewer's Responses to Questions

**Comments to the Author**

1. Is the manuscript technically sound, and do the data support the conclusions?

Reviewer #1: Yes

Reviewer #2: No

2. Has the statistical analysis been performed appropriately and rigorously? 

Reviewer #1: I Don't Know

Reviewer #2: No

3. Have the authors made all data underlying the findings in their manuscript fully available?

Reviewer #1: Yes

Reviewer #2: Yes

4. Is the manuscript presented in an intelligible fashion and written in standard English?

Reviewer #1: Yes

Reviewer #2: Yes

5. Review Comments to the Author

Reviewer #1: This is my review of the manuscript, entitled “Seeing through the forest: The gaze path to purchase.” The authors present an eye-tracking study on 92 participants examining, in a 3 (display size) x 2 (price) within-subjects design, the link between the gaze sequence in which individuals viewed a display and their hypothetical choice of an unbranded plant product. While I am not able to evaluate the authors’ machine learning approach, my expertise lies in eye-tracking research, when applied to business settings. In that sense, I enjoyed reading the present manuscript, and I think the authors have a well-crafted initial submission. Therefore, based on my feedback provided below, my reviewer recommendation is to invite the authors for a minor revision. I start by delineating the larger topical concerns and areas for improvements, after which I continue with some minor suggestions to improve the readability and contribution of this piece.

MAJOR POINTS:

• In connection to the authors’ theorizing and discussion on gaze aversion, I was missing some notes on whether and how this construct is related to, or clearly distinct from peripheral vision, which plays an important role in consumer choice situations. For example, how does the gaze aversion conceptualization differ from the account proposed by Wästlund et al. (2018; references provided in the end of this review)? This is important given that the authors write at several places that consumers often ignore some visual cues.

• While eye-tracking studies in lab settings have their merits, they lack a certain degree of ecological validity. This should be acknowledged in the limitations section of the authors’ manuscript.

• I found the authors’ exposition to be a bit front-heavy, particularly the review of literature section, which I think can be written more succinctly.

• When it comes to the authors’ contribution and significance statements, I think they should make more of an effort in order to not over-sell their findings. For example, they note that “few of the studies investigated the gaze paths in which visual cues were accessed and no studies investigated whether that relationship predicted purchase” (p. 4, last paragraph); that their “goal is to better understand the sequence of cues that help get products off the shelf and into the cart” (p. 5, first paragraph); and imply that they look at actual product choices and purchase decisions. Strictly speaking, however, all these claims are inaccurate, as the authors are solely examining hypothetical choice in a rather artificial lab setting (see my ecological validity point above). Please be precise.

• I think the authors should report a 3 (display) x 2 (price) within-subjects ANOVA on the 11-point likelihood of buying scale, both as a validity check and as a way to potentially boost their managerial implications. Certainly, it would be interesting for a retail owner to know if there is a main effect of display size, such that, for instance, people indicate greater buying likelihood in a display with a larger (vs. smaller) number of product options. Similarly, it would be interesting, as a validity check, to see if participants’ buying likelihood were greater in the presence (vs. absence) of price discounts (with corresponding effect sizes), and to explore whether display size interacted with price to influence participants’ buying likelihood. Please implement these analyses or provide very solid arguments as for why such details are not included (arguments of the type “fell beyond the purpose of our study” is not enough in my view, but I will not hijack the story).

• The authors provide arguments supporting their findings on product expertise and previous annual purchases of plants. However, virtually no arguments at all appear for the finding that race (white) helped improve the model’s predictive accuracy, which left me with the question “why?” Please elaborate.

MINOR POINTS:

• Please specify the design (within-subjects) in the abstract.

• Please correct the claimed non-student sample in the abstract, as the main text in the manuscript seems to suggest that some students were also included in the study.

• For the sales figures presented on p. 2 (final paragraph), the authors may want to have a more recent reference than 2016. Relatedly, they may want to add the fact that retail sales from physical stores still account for 90-95% of total retail sales (e.g., Otterbring & Lu, 2018).

• At p. 4 (first paragraph), I got confused, as the authors start by stating that the number of visual cues at the POP (point-of-purchase) can be OVERWHELMING at times, after which they state that their main goal is to study the relationship between gaze sequences and product choice for MINIMALLY packaged, UNBRANDED products.

• When reviewing visual attention and purchase likelihood, the authors may want to add some recent eye-tracking references using hypothetical choice as the focal dependent variable (e.g., Ballco et al., 2019; Otterbring et al., 2020; Otterbring & Shams, 2019; Tórtora et al., 2019).

• At p. 6 (first paragraph), the authors indicate some inconsistencies in findings, but they do not specify how these inconsistencies are manifested. Please clarify.

• The real, imagined, or implied presence of others tend to influence individuals’ visual attention (e.g., Lie & Yu, 2017; Milani et al., 2019; Wong & Stephen, 2019). Therefore, I think further methodological details are needed in the manuscript with respect to whether an experimenter was present in the same room as the participant(s) during the study and how many individuals that attended the eye-tracking sessions simultaneously.

• Did all the signs displayed in the study use the same color scheme on and around the letters?

• Please clearly indicate the sampling frequency and accuracy of the eye-tracking device used.

• At p. 16 (last paragraph), the authors state that CEN was identical for one model and lower for four of the five remaining models after adding the consumer attributes, after which they state that overall accuracy was identical with and without the addition of consumer data for two of the six models. This seems contradictory – was it one or two models that were not improved after addition of the consumer attributes? Please clarify.

• Please present the zero-order correlations between the included consumer attributes: expertise, being (vs. not being) white, and prior plant purchase.

In sum, I think the authors have an interesting study and my evaluation as a reviewer is that they may have the chance of getting their work published, in case they meticulously address all points raised above. I wish them good luck in their future work on this topic and I hope that my comments were helpful.

References

Ballco, P., de-Magistris, T., & Caputo, V. (2019). Consumer preferences for nutritional claims: An exploration of attention and choice based on an eye-tracking choice experiment. Food Research International, 116, 37-48.

Liu, N., & Yu, R. (2017). Influence of social presence on eye movements in visual search tasks. Ergonomics, 60(12), 1667-1681.

Milani, S., Brotto, L. A., & Kingstone, A. (2019). “I can see you”: The impact of implied social presence on visual attention to erotic and neutral stimuli in men and women. The Canadian Journal of Human Sexuality, 28(2), 105-119.

Otterbring, T., Gidlöf, K., Rolschau, K., & Shams, P. (2020). Cereal Deal: How the Physical Appearance of Others Affects Attention to Healthy Foods. Perspectives on Behavior Science, 1-18.

Otterbring, T., & Lu, C. (2018). Clothes, condoms, and customer satisfaction: The effect of employee mere presence on customer satisfaction depends on the shopping situation. Psychology & Marketing, 35(6), 454-462.

Otterbring, T., & Shams, P. (2019). Mirror, mirror, on the menu: Visual reminders of overweight stimulate healthier meal choices. Journal of Retailing and Consumer Services, 47, 177-183.

Wästlund, E., Shams, P., & Otterbring, T. (2018). Unsold is unseen… or is it? Examining the role of peripheral vision in the consumer choice process using eye-tracking methodology. Appetite, 120, 49-56.

Tórtora, G., Machín, L., & Ares, G. (2019). Influence of nutritional warnings and other label features on consumers' choice: Results from an eye-tracking study. Food Research International, 119, 605-611.

Wong, H. K., & Stephen, I. D. (2019). Eye tracker as an implied social presence: awareness of being eye-tracked induces social-norm-based looking behaviour. Journal of Eye Movement Research, 12(2), 1-17.

Reviewer #2: This study investigates the relation between overt attention and purchase choices (i.e., buying a plant) in a real-world scenario using 3D Tobii glasses. Participants’ gaze positions are monitored while asked to choose one of plant out of different many, and each is tested in six different conditions, which result from a manipulation of set size (6, 12, 24 plants), and the price band (low, high). From the eye-movement responses, they seem to consider the frequency of the first three fixations, and together with survey data of their consumer preferences, train a set of different random forest classifiers aimed at predicting which plant will be chosen. The classifiers seem to display a good accuracy for the small set-size (72%) and their performance improve when trained with survey data and a feature of the eye-movement defined as Look Away to Decide (LATD), which represents fixations outside of any Area of Interest.

It is interesting to read about the application of eye-tracking technology in different disciplines, and I truly appreciate the effort of building computational models based on eye-movement responses to make predictions in real-life contexts. The authors have also provided the data and scripts to run the analyses which is a nice and honourable aim. However, overall, the study falls short on its methodology, theoretical underpinnings and implications for it to make a substantial contribution.

In what follows, I highlight my major concerns and a few minor other comments that the authors may want to take on board to improve their study.

Major comments:

(a) There seems to be confusion about what the theoretical contributions of this work are and especially what their novelty should be. Since the seminal work of Yarbus, 1967, it is widely known that eye-movement responses can successfully predict the task participants perform. The notion of task can be broadly construed to encompass several things among which also the target object that people may be attentively selecting. It is not surprising then, that when considering the first three fixations only (more later about why this may be a methodologically problematic choice) it may be enough to predict what object is receiving most attention, and so correlate with the final choice. So, I find this result, per-se, not novel nor striking. What’s also odd is that the authors claim that looking away from the object is a good predictor of choice, that’s counterintuitive as usually fixation time correlates with the amount of attention that a target got. Furthermore, even in the context of consumer psychology, there are other studies, some of which mentioned by the authors, showing already systematic relations between gaze behaviour and choice.

(b) The other novelty seems to be that the inclusion of LATD and survey data improves the prediction accuracy of the models. However, the predictions of the models presented in Table 1 seem to be erratic across display types, and do not follow any clear theoretical logic as to why that’s the case. In fact, the best accuracy seems to be Model 3 for a display of 6 plants at low price (called J, 78%), which includes consumer attributes. At the same time, we also see that a display with 24 plants and a high price (Q) reaches an accuracy of 73% with Model 2, which is the one with LATD features but not consumer attributes. If both consumer attributes and LATD are included as training features (Model 4), we see an incredible drop in performance for the display Q (36%), while for the display P, also 24 plants and low-price, there is a much better accuracy (75%). The lack of consistency across displays, or between training data and achieved model performance casts some important doubts about the reliability of these classification results, so undermining any theoretical statements related to it.

(c) LATD feature is tentatively argued to be reflecting gaze aversion. But, instead, it could simply relate to data quality and nothing else. In fact, it does reflect fixations falling outside all possible area of interests. I would be really surprised that participants actively engaged in a choice task would spend time looking at no informative areas of the display. Thus, it would be important to quantify how many fixations fall outside the area of interests, whether the percentage of such fixations varies (or it is stable) across participants and displays, etc. In practice, shed better light to the nature of such out of range fixations. In general, there is very little detail about eye-tracking data quality, which is much needed. Also, the argument of gaze aversion does not really fly, in my humble opinion, as the literature this hypothesis refers to involves human-human interaction, while here, it is a very different setting with no-animate agents taking parts beyond the participants.

(c) The study needs to be much better contextualised within the visual cognition literature rather than only using references to consumer psychology. There are few claims or arguments about bottom-up vs. top-down processes, for example, that seem misleading, inaccurate or poorly referenced. I have added a few works on the topic at the end of this review, so that they may give the authors a starting lead to relevant literature. Line 106:108 “Attention drawn to a sudden loud noise, flashing lights, or an attractive person is the result of a bottom-up process, while finding a friend’s face in a crowd is the result of a top-down process.” There should be a mention of what drives bottom-up processes, e.g., low-level visual saliency, and clarify what is meant by top-down. Is it the role of the task, e.g., visual search as opposed to free-viewing? Why is looking for faces top-down, because of the stimuli (i.e., faces) or because of the task? Also, it is not clear how the current study taps into such a dichotomy. In fact, as far as I can see, the experimental design does not systematically manipulate any of the two types of mechanisms. There is a manipulation of set size, referred by the authors as displays, and price, but none of the two is systematically related to any of the results presented. As said in my previous context, the results from the modelling do not show any systematicity with respects to these two independent variables.

(d) There is also a long-standing literature about eye-movement scan-patterns, starting from the seminal paper by Noton & Stark 1971, and about modelling, or measuring similarity, between them. The authors seem not to be fully aware about this bulk of literature, and I think instead that such literature should be better discussed. Also, their analysis of scan-patterns sounds a bit unusual. For what I have understood, from all possible fixations that each participant made during a trial, only the first three were kept, and they used a rather unconventional naming for it (i.e., 3-mers). There is a weak attempt to explain why that was done but it did not fully convince me. I think the authors should show what happened when the scan-patterns were fully considered, or at least show, what happens to the modelling results as they increase the number of data-points from 3 to N. More importantly, from their analysis I gathered that the sequential aspect of scan-patterns was not considered. It seems that the random forest classifiers were in fact trained using relative frequencies of fixating at the different AOI, if that’s the case, then I do not see how this data can be defined as a scan-pattern. Modelling a scan-pattern implies preserving its sequential aspect, and utilising methods that can maintain it untouched, e.g., Hidden Markov Models. They should perhaps refocus their analyses in terms of extracting eye-movement features, and that they decided to consider the first three fixations as those one carrying important information about the task. I also wonder whether they may look for other characteristics of the oculo-motor behaviour, it may well be that there are other types of features, object-based, best fit to their classification task.

(e) The justification of using Random Forest instead of any other methods seems a bit weak. The authors say, Line: 289-291: “We selected the RF machine learning approach because of its many properties for classifying a high-dimensional data set such as the present one, which would be impractical in the context of traditional linear models due to parameter estimation problems.” I am not sure why, for example, Support-Vector machine would not work for their purpose. SVM can be trained on a technically infinite high-dimensional feature space to predict a certain outcome variable. As the features their random forest classifiers were trained on are relative frequencies of looks to AOI and survey data, I guess that SVM would also here. I think, they should better motivate their choice, and show that their method of classification, i.e., random forest, would really give them a better F-score than some other method. By cross validating their predictions using different methods may help them also realise what may be generating variability across displays.

(f) Survey data is used to train the random forests, but it is never explained at length what is the data consisting of, how is it treated, and inputted. For example, I take that each participant contributed to a single data point per questionnaire, but there are many more eye-movement data points per display. It is crucial that the treatment of the data is much better elucidated, otherwise the reader cannot really follow and appreciate what was done in the modelling.

(g) The task itself is severely underspecified. I am still not sure, even after reading the paper a few times, whether participants had always to make a choice to buy something or not. If not, it is not clear how many times they did not make any choice, and how were these trials treated. In general, it is never clear how many trials were used to train the classifiers. Even if the data is provided, there is the need for greater details about the treatment of the data, and better descriptive statistics.

Minor comments:

- It may be useful to give an example of the Area of Interests. There seems to be a lot of overlap between the plants in the display, so I wonder how boundaries are defined. Also, the plants further away in the display may receive less fixations just because of their relative position.

- Improve the quality of the only Figure, which is rather low. Also, it is not clear what this Figure is depicting. I would suggest to add more Figures to aid the understanding of the readers.

- Line 283:284 “The next step was to investigate whether gaze 3-mers would be predictive of product choice. For that purpose, we computed, for each participant, the relative frequency in which each 3-mer occurred in the first 100 gaze events in their gaze sequence.” Not sure I understand this passage. I got that 3-mers are the first 3 fixations in a sequence, how can they be 100 gaze events? How did they derive this number? Are 3-mers, three area of interest? If so, how were these regions chosen if the display contains many more (i.e., a minimum of 7 and a maximum of 25 including out of range fixations)?

- Line 256:257 The authors mention a chocolate bars practice task, but it is never clear why was this done, and what did it entail.

- Line 258:259. The participants have to verbally state how much from 0 to 10 would they like to buy a product. But, was not it a buy vs. no-buy binomial outcome variable that’s modelled? If it is a continuous variable, what are the random forest predicting? A preference score? If so, how is prediction accuracy then measured?

- Line 231:232 “Plants occupied approximately 0.12 m2, 0.25 m2, and 0.45 m2 of display space for the 6, 12, and 24-plant designs, respectively” The authors should provide measurements in terms of degree of visual angle, as the size of objects is relative to the distance of the observer.

Line 202: There seems to be a typo.

Line 309:312 “Our final models focused on predicting consumer choice for plants selected by at least 10 consumers (two plants in six-plant displays and three plants in the 12- and 24-plant displays) and for the no-plant choice.” Obscure passage. This sounds like data dredging. It is unclear how many final data points contributed to the modelling.

Line 323:324 “To determine which attributes had a strong impact on the plant choice, variable selection was performed using a series of F-tests.” The authors now mention feature selection, but it was never clear how many features were originally selected, and what were F-tests based on.

Table 1: There is what seems an arbitrary list of letters to identify the displays. Not sure what a reader could make of it.

Line 364: There is a mention of baseline, but it is not clear what the baseline is, and how it is computed.

Line 400:403 There is a mention to EEG research but it feels quite tangential and off-topic.

References

Coco, M. I., & Keller, F. (2014). Classification of visual and linguistic tasks using eye-movement features. Journal of Vision, 14(3), 11. https://doi.org/10.1167/14.3.11

Itti, L., Koch, C., & Niebur, E. (1998). A model of saliency-based visual attention for rapid scene analysis. IEEE Transaction on Pattern Analysis and Machine Learning, 20(11), 1254–1259. https://doi.org/10.1109/TPAMI.2012.125

Noton, D., & Stark, L. (1971). Scanpaths in eye movements during pattern perception. Science, 171(3968), 308–311. https://doi.org/10.1126/science.171.3968.308

Rothkopf, C. A., Ballard, D. H., & Hayhoe, M. M. (2007). Task and context determine where you look. Journal of Vision, 1410, 1–20.

Simola, J., Salojärvi, J., & Kojo, I. (2008). Using hidden Markov model to uncover processing states from eye movements in information search tasks. Cognitive Systems Research, 9(4), 237–251. https://doi.org/10.1016/j.cogsys.2008.01.002

Treisman, A. M., & Gelade, G. (1980). A feature-integration theory of attention. Cognitive Psychology, 12(1), 97–136. https://doi.org/10.1016/0010-0285(80)90005-5

Yarbus, A. L. (1967). Eye Movements and Vision. Plenum Press.

Wolfe, J. M., & Utochkin, I. S. (2019). What is a preattentive feature? Current Opinion in Psychology, 29, 19–26. https://doi.org/10.1016/j.copsyc.2018.11.005

6. PLOS authors have the option to publish the peer review history of their article (what does this mean?). If published, this will include your full peer review and any attached files.

Reviewer #1: No

Reviewer #2: No

---

## [Author Response · Author response to Decision Letter 0]

26 Aug 2020

We have responded to all reviewer comments in the file labeled "Response to reviewers". Thank you.

---

## [Decision Letter · Decision Letter 1]

22 Sep 2020

Seeing through the forest: The gaze path to purchase

PONE-D-20-14107R1

Dear Dr. Behe,

We’re pleased to inform you that your manuscript has been judged scientifically suitable for publication and will be formally accepted for publication once it meets all outstanding technical requirements.

Kind regards,

Claudio Soregaroli

Academic Editor

PLOS ONE

Additional Editor Comments (optional):

Reviewers' comments:

Reviewer's Responses to Questions

**Comments to the Author**

1. If the authors have adequately addressed your comments raised in a previous round of review and you feel that this manuscript is now acceptable for publication, you may indicate that here to bypass the “Comments to the Author” section, enter your conflict of interest statement in the “Confidential to Editor” section, and submit your "Accept" recommendation.

Reviewer #1: All comments have been addressed

Reviewer #2: All comments have been addressed

2. Is the manuscript technically sound, and do the data support the conclusions?

Reviewer #1: Yes

Reviewer #2: Partly

3. Has the statistical analysis been performed appropriately and rigorously? 

Reviewer #1: Yes

Reviewer #2: Yes

4. Have the authors made all data underlying the findings in their manuscript fully available?

Reviewer #1: Yes

Reviewer #2: Yes

5. Is the manuscript presented in an intelligible fashion and written in standard English?

Reviewer #1: Yes

Reviewer #2: Yes

6. Review Comments to the Author

Reviewer #1: This is a responsive revision. I have no further comments and I recommend acceptance of the manuscript.

Reviewer #2: The authors have done a very thorough and careful job at addressing my major concerns. I feel that the addition of the new SVM analysis, and especially the clarification of the k-mers fixation sequences as rendered the paper much clearer, hence strengthening its theoretical contributions.

7. PLOS authors have the option to publish the peer review history of their article (what does this mean?). If published, this will include your full peer review and any attached files.

Reviewer #1: No

Reviewer #2: **Yes: **Moreno I. Coco

---

## [Editor Report · Acceptance letter]

25 Sep 2020

PONE-D-20-14107R1 

Seeing through the forest: The gaze path to purchase 

Dear Dr. Behe:

I'm pleased to inform you that your manuscript has been deemed suitable for publication in PLOS ONE. Congratulations! Your manuscript is now with our production department. 

Kind regards, 

on behalf of

Dr. Claudio Soregaroli 

Academic Editor

PLOS ONE